# Language-Driven Active Learning for Diverse Open-Set 3D Object Detection

## Abstract

*Object detection is crucial for ensuring safe autonomous driving. However, data-driven approaches face challenges when encountering minority or novel objects in the 3D driving scene. In this paper, we propose VisLED, a language-driven active learning framework for diverse open-set 3D Object Detection. Our method leverages active learning techniques to query diverse and informative data samples from an unlabeled pool, enhancing the model's ability to detect underrepresented or novel objects. Specifically, we introduce the Vision-Language Embedding Diversity Querying (VisLED-Querying) algorithm, which operates in both open-world exploring and closed-world mining settings. In open-world exploring, VisLED-Querying selects data points most novel relative to existing data, while in closed-world mining, it mines new instances of known classes. We evaluate our approach on the nuScenes dataset and demonstrate its effectiveness compared to random sampling and entropy-querying methods. Our results show that VisLED-Querying consistently outperforms random sampling and offers competitive performance compared to entropy-querying despite the latter's model-optimality, highlighting the potential of VisLED for improving object detection in autonomous driving scenarios. We make our code publicly available at [anonymized].*

## 1. Introduction

Object detection is critical for safe autonomous driving. Data-driven approaches currently provide the best performance in detecting and localizing objects in the 3D driving scene. Detection models perform best on objects which are most represented in driving datasets. This creates challenges when some objects are less represented (minority classes), or unrepresented within the annotation scheme ("novel" objects [1], relevant for "open-set" learning [2]), and becomes especially important when minority objects are most salient to driving decisions [3–6]. Further, from a pragmatic standpoint, the collection, curation, and annotation of such datasets can be extremely expensive [7, 8],

motivating the use of heuristics and algorithms which limit annotation efforts while maximizing model learning.

## 2. Related Research

Active learning methods are driven by a query function which selects relevant data from an unlabeled pool to be annotated and joined to the training set. These methods broadly divide into two classes: uncertainty-based and diversity-based methods [9]. In uncertainty-based methods, data is selected by the query function's assessment of how confusing the datum is *to the existing model*. On the other hand, in diversity-based methods, data is selected by being distinct from existing training data by some measure, and this can be done without consideration of the learning model.

### 2.1. The Role of Uncertainty and Diversity-Based Methods in Closed and Open Set Learning

In closed-set learning, it is assumed that a system should classify or learn about a fixed set of target classes. By contrast, in open-set learning, the system assumes that it may encounter novel data which belongs to a class unrepresented by its current target set. Naturally, this brings up many research challenges in recognizing this novelty when it appears, determining when to define a new set construct, and integrating new constructs into the learning mechanism.

Here, we suggest that diversity-based methods are particularly well-suited for these open-set learning tasks. Because uncertainty-based methods select relative to their existing world model, there is an inductive bias imposed in relating new data to existing patterns. On the other hand, in diversity-based methods, data is compared only to other data, analogous to unsupervised learning. This does create a tradeoff: closed-set learning excels under uncertainty-driven sampling, since these methods are optimized for the current world model and target set, but cannot treat the world as "open" as diversity-driven sampling. But, critically, we show in this research that diversity-based active learning still provides a benefit to the learning system (even if not "optimal" to the particular model and set definition), *and* is suitable for open-set data selection.

## 2.2. Learning from Vision-Language Representations

Prior research has shown that vision-language representations such as embeddings from contrastive language-image pretraining (CLIP) [10] can be used to identify novelty of an image relative to a set (and, as a bonus, can be decoded into a verbal explanation of novelty) [11]. In our research, we utilize this representation and corresponding ability to select novel images as a proxy for the amount of useful, previously-unexplored information within a complete multimodal driving scene, allowing for an active learning query to select diverse samples based on vision-language encodings of scene images.

## 3. Algorithm

Here, we present our algorithm named Vision-Language Embedding Diversity Querying (VisLED-Querying), which can be viewed in Figure 1. The algorithm can be used in two different settings:

1. Open-World Exploring: this method imposes no particular class expectations on the data. It is suitable for cases when the model seeks to include information which is most novel relative to data it has seen previously.

2. Closed-World Mining: this method utilizes a zero-shot learning [10] step to sort data between a fixed set of classes before evaluating for novelty, filtering any points estimated to not belong to one of the closed-set classes. This method is suitable for mining new and different instances of existing classes, but may also filter out the most difficult or unusual instances even from known classes if the zero-shot method fails to recognize the object.

---

**Algorithm 1:** Open-World Exploring VisLED-Querying

---

**Input:** Unlabeled pool of egocentric driving scene images
**Output:** Updated training set
Embed each egocentric driving scene image from the unlabeled pool using CLIP;
Use hierarchical clustering to separate the embeddings;
Sample new data points from the unclustered set for addition to the training set;

---

When employing CLIP's [12] zero-shot learning technique for classification, the algorithm examines each sample image to identify objects, that are most likely to belong to predefined classes. As a result, each sample is assigned to a single class, as the zero-shot learning method predominantly identifies one class with high accuracy. In instances where other classes may also be identified, their confidence scores are typically low enough to risk false positives, rendering them inadequate for threshold-based classification. Therefore, a single-class assignment is favored for simplicity and accuracy.

Once the samples for each class have been identified, embeddings will be generated separately for each class, followed by hierarchical clustering. Subsequently, a number of samples will be selected from each class, with a focus on sampling from clusters with minimal data representation. Initially, the algorithm will prioritize unique samples (clusters with only one sample present), matching them with corresponding scene names until the desired number of unique scenes is achieved in the training set. Upon inclusion of all scene-names from unique samples, the algorithm will proceed to clusters containing pairs of images, and so on, until the required number of scenes have been sampled for the training set.

---

**Algorithm 2:** Closed-World Mining VisLED-Querying

---

**Input:** Unlabeled pool of egocentric driving scene images
**Output:** Updated training set
Embed each egocentric driving scene image from the unlabeled pool using CLIP;
Encode each class label using a text encoding;
Applying zero-shot learning by maximizing the product of the embeddings, sort the embedded images by class;
For each class, apply hierarchical clustering;
Sample new data points from the unclustered set associated with the desired class, and add to the training set;

---

## 4. Experimental Evaluation

### 4.1. Dataset

We use the nuScenes object detection dataset [13] for our experiments. nuScenes contains 1.4M camera images and 400k LIDAR sweeps of driving data, originally labeled by expert annotators from an annotation partner. 1.4M objects are labeled with a 3D bounding box, semantic category (among 23 classes), and additional attributes. NuScenes comprises 1000 scenes. In order to maintain complete control over the scenes within the dataset, we modify the fundamental database setup slightly, using the method introduced in [14, 15] to accommodate active learning queries. We use the *trainval* split of the dataset for public reproducibility.

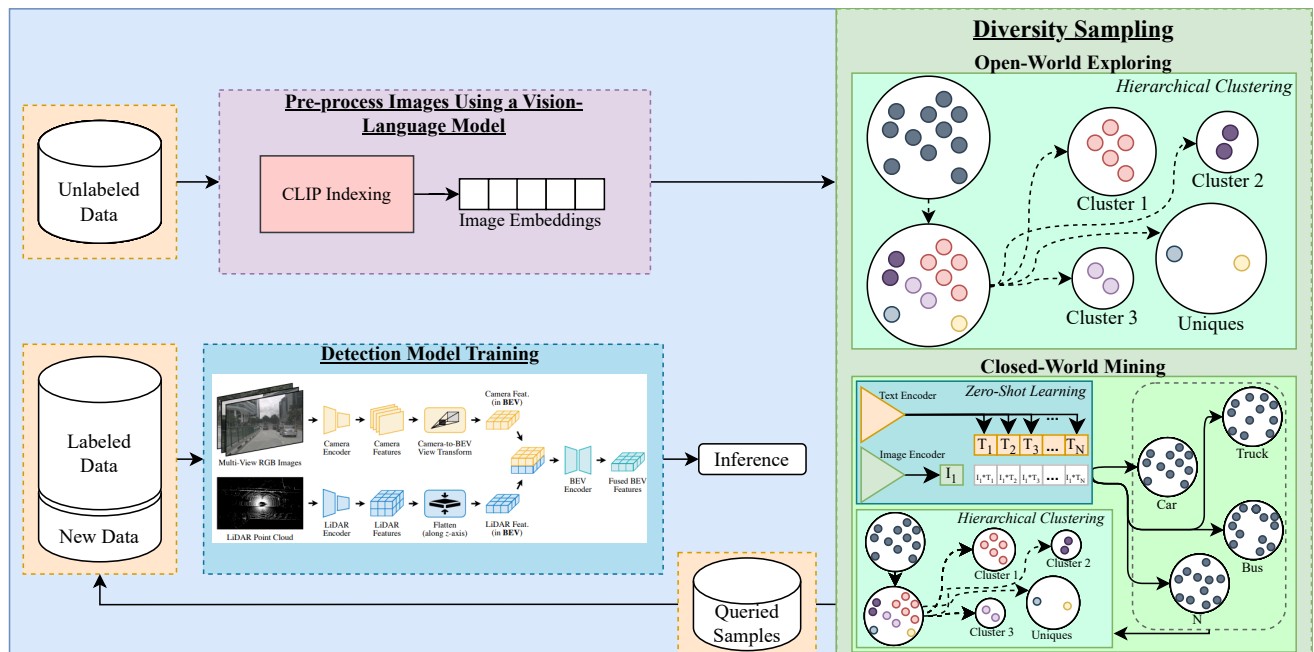

Figure 1. VisLED System Overview. For both Open-World Exploring and Closed-World Mining, the system begins with the processing of the unlabeled data pool into vision-language embedding representations. In Open-World Exploring, these embeddings are clustered and used as the basis for a query. In Closed-World Mining, the embeddings are first used in zero-shot learning to classify scenes based on object appearance, and then further clustered per-class, offering a chance to sample from particular classes which are known to be minority in the labeled training set.

## 4.2. 3D Object Detection Model

We explore the BEVFusion approach to 3D object detection [16], which has demonstrated notable performance, ranking third in the NuScenes tracking challenge and seventh in the detection challenge. While various methods exist to integrate image and LiDAR data into a unified representation, LiDAR-to-Camera projection methods often introduce geometric distortions, and Camera-to-LiDAR projections face challenges in semantic-orientation tasks. BEVFusion aims to address these issues by creating a unified representation that preserves both geometric structure and semantic density.

In our implementation, we utilize the Swin-Transformer [17] as the image backbone and VoxelNet [18] as the LiDAR backbone. To generate bird's-eye-view (BEV) features for images, we employ a Feature Pyramid Network (FPN) [19] to fuse multi-scale camera features, resulting in a feature map one-eighth of the original size. Subsequently, images are down-sampled to 256x704 pixels, and LiDAR point clouds are voxelized to 0.075 meters to obtain the BEV features necessary for object detection. These modalities are integrated using a convolution-based BEV encoder to mitigate local misalignment between LiDAR-BEV and camera-BEV features, particularly in scenarios of depth estimation uncertainty from the camera mode. For a compre-

hensive overview of the architecture, including its integration with VisLED-Querying, refer to Figure 1.

## 4.3. Experiments

We train the BEVFusion model in increasing training set sizes, using three different acquisition modes: (1) Random Sampling, (2) Entropy-Querying, and (3) VisLED-Querying with Closed-Set Mining setting. As expected, active learning strategies outperform the random baseline, and the entropy-querying method is dominant due to its nature of optimizing uncertainty with respect to the model, as opposed to VisLED's model-agnostic sampling. Yet, as illustrated in Table 1, VisLED still stays consistently ahead of random sampling, and offers a 1% gain over random sampling mAP at 50% of the data pool, all without requiring *any* model training or inference.

## 5. Discussion and Conclusion

Our presented learning method, VisLED-Querying, samples without any information about the model. This enables VisLED to select novel, informative data points, to the extent that novelty is visibly identifiable, for *any* model. The benefit this offers is that a data point may need to be annotated only once, and can then be used in a variety of models for additional autonomous driving tasks instead of

| Labeled Pool | | mAP | | | NDS | | |
| --- | --- | --- | --- | --- | --- | --- | --- |
| Rounds | % | Random | Entropy | VisLED | Random | Entropy | VisLED |
| 1 | 10% | 30.95 | 31.06 (+1.06) | 29.14 (-1.81) | 33.53 | 34.09 (+0.56) | 32.16 (-1.37) |
| 2 | 20% | 38.00 | 40.41 (+2.41) | 40.76 (+2.76) | 40.14 | 41.85 (+1.71) | 41.18 (+1.04) |
| 3 | 30% | 44.94 | 45.57 (+0.63) | 45.01 (+0.07) | 48.41 | 50.11 (+1.7) | 49.40 (+0.99) |
| 4 | 40% | 47.73 | 49.24 (+1.51) | 49.21 (+1.48) | 53.10 | 53.80 (+0.7) | 53.64 (+0.54) |
| 5 | 50% | **49.90** | 63.88 (+13.98) | **51.05** (+1.15) | **55.64** | 64.85 (+9.21) | **56.45** (+0.81) |
| | 100% | 52.88 | | | 58.73 | | |

Table 1. This table shows the mean average precision (mAP) and nuScenes driving score (NDS) metrics for the random sampling, entropy-querying, and VisLED-querying (Closed-World Mining) in every round. It also shows the mAP and NDS scores for the full training split when trained using one GPU. Both the entropy-querying and VisLED methods outperform random sampling consistently, and reach nearly the same level of performance as 100% of the data at just the 50% data point, showing faster learning than the baseline method.

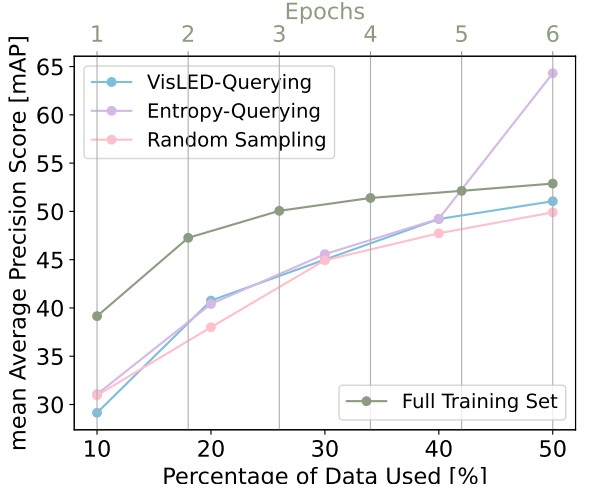
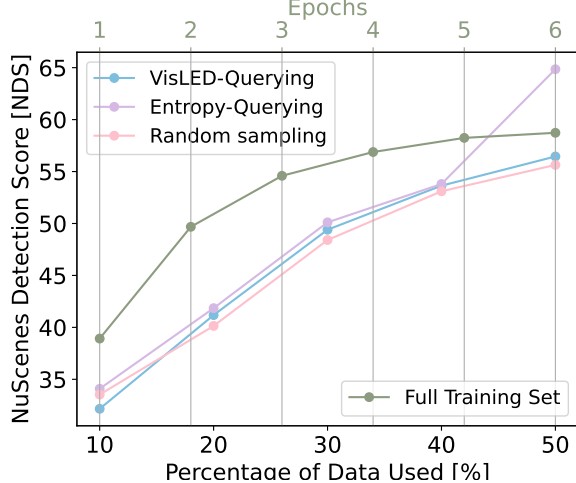

Figure 2. Performance of BEVFusion in 3D Object Detection on nuScenes at different training set sizes, using three different learning strategies. Simultaneously, we chart the learning of BEVFusion on the full training set, over the course of six epochs (top horizontal axis) to give an impression of the asymptotic performance limit that may be expected of the model. We observe that the active learning methods move towards this asymptote sooner than random sampling, and that VisLED maintains a margin over random sampling throughout.

sampling and possibly forming an entirely different set for annotation. While these gains may be marginal in the current data setting ($< 1000$ scenes), at scale, these performance gains may translate to serious reductions in annotation costs and safety-critical detection failures. Further, VisLED offers one key possibility that is otherwise limited on uncertainty-driven approaches: VisLED will recommend unique samples without any prior assumptions on class taxonomy, making it especially suited to open-set learning, where new classes may be introduced at any time. This capability, when paired with methods of self- or semi-supervised learning for object detection by fusing LiDAR and camera [20], may prove especially beneficial in identifying and learning from novel encounters. In future research, we plan to experiment on the effectiveness of VisLED in multi-task learning settings [21], experiments on other benchmark datasets [22], and experiments in open-set and continual learning.

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
