# OpenReview forum: "Language-Driven Active Learning for Diverse Open-Set 3D Object Detection"
_thecvf.com/CVPR/2024/Workshop/VLADR — VLADR 2024 Poster_

### Official Review · Reviewer_eFAA · 2024-04-17
**Reviews**

**Rating:** 6
**Confidence:** 4

**Review:**

This paper presents an approach to effectively mine scenes of interest from a large pool of unlabeled ego-centric driving logs. Leveraging CLIP's embeddings is a common but still clever way to handle open-vocabulary querying settings.
It's highly relevant to our workshop. I tend to accept this paper, but I also suggest authors to further polish the paper's presentation.

---

### Decision · Program_Chairs · 2024-04-22

Accept (Poster)